# AN OPEN REVIEW OF OPENREVIEW: A CRITICAL ANALYSIS OF THE MACHINE LEARNING CONFERENCE REVIEW PROCESS

## ABSTRACT

Mainstream machine learning conferences have seen a dramatic increase in the number of participants, along with a growing range of perspectives, in recent years. Members of the machine learning community are likely to overhear allegations ranging from randomness of acceptance decisions to institutional bias. In this work, we critically analyze the review process through a comprehensive study of papers submitted to ICLR between 2017 and 2020. We quantify reproducibility/randomness in review scores and acceptance decisions, and examine whether scores correlate with paper impact. Our findings suggest strong institutional bias in accept/reject decisions, even after controlling for paper quality. Furthermore, we find evidence for a gender gap, with female authors receiving lower scores, lower acceptance rates, and fewer citations per paper than their male counterparts. We conclude our work with recommendations for future conference organizers.

## 1 INTRODUCTION

Over the last decade, mainstream machine learning conferences have been strained by a deluge of conference paper submissions. At ICLR, for example, the number of submissions has grown by an order of magnitude within the last 5 years alone. Furthermore, the influx of researchers from disparate fields has led to a diverse range of perspectives and opinions that often conflict when it comes to reviewing and accepting papers. This has created an environment where the legitimacy and randomness of the review process is a common topic of discussion. Do conference reviews consistently identify high quality work? Or has review degenerated into a process orthogonal to meritocracy?

In this paper, we put the review process under a microscope using publicly available data from across the web, in addition to hand-curated datasets. Our goals are to:

- **Quantify reproducibility in the review process** We employ statistical methods to disentangle sources of randomness in the review process. Using Monte-Carlo simulations, we quantify the level of outcome reproducibility. Simulations indicate that randomness is not effectively mitigated by recruiting more reviewers.

- **Measure whether high-impact papers score better** We see that review scores are only weakly correlated with citation impact.

- **Determine whether the process has gotten worse over time** We present empirical evidence that the level of reproducibility of decisions, correlation between reviewer scores and impact, and consensus among reviewers has decreased over time.

- **Identify institutional bias** We find strong evidence that area chair decisions are impacted by institutional name-brands. ACs are more likely to accept papers from prestigious institutions (even when controlling for reviewer scores), and papers from more recognizable authors are more likely to be accepted as well.

- **Present evidence for a gender gap in the review process** We find that women tend to receive lower scores than men, and have a lower acceptance rate overall (even after controlling for differences in the topic distribution for men and women).

## 2 DATASET CONSTRUCTION

We scraped data from multiple sources to enable analysis of many factors in the review process. *OpenReview* was a primary source of data, and we collected titles, abstracts, authors lists, emails, scores, and reviews for ICLR papers from 2017-2020. We also communicated with OpenReview maintainers to obtain information on withdrawn papers. There were a total of 5569 ICLR submissions from these years: ICLR 2020 had 2560 submissions, ICLR 2019 had 1565, ICLR 2018 had 960, and ICLR 2017 had 490. Authors were associated with institutions using both author profiles from OpenReview and the open-source World University and Domains dataset. *CS Rankings* was chosen to rank academic institutions because it includes institutions from outside the US.

The *arXiv* repository was scraped to find papers that first appeared in non-anonymous form before review. We were able to find 3196 papers on arXiv, 1415 of them from 2020.

Citation and impact measures were obtained from *SemanticScholar*. This includes citations for individual papers and individual authors, in addition to the publication counts of each author. SemanticScholar search results were screened for duplicate publications using an edit distance metric. The dataset was hand-curated to resolve a number of difficulties, including finding missing authors whose names appear differently in different venues, checking for duplicate author pages that might corrupt citation counts, and hand-checking for duplicate papers when titles/authors are similar.

To study gender disparities in the review process, we produced gender labels for first and last authors on papers in 2020. We assigned labels based on gendered pronouns appearing on personal webpages when possible, and on the use of canonically gendered names otherwise. This produced labels for 2527 out of 2560 papers. We acknowledge the inaccuracies and complexities inherent in labeling complex attributes like gender, and its imbrication with race. However we do not think these complexities should prevent the impact of gender on reviewer scores from being studied.

### 2.1 TOPIC BREAKDOWN

To study how review statistics vary by subject, and control for topic distribution in the analysis below, we define a rough categorization of papers to common ML topics. To keep things simple and interpretable, we define a short list of hand curated keywords for each topic, and identify a paper with that topic if it contains at least one of the relevant keywords. The topics used were `theory`, `computer vision`, `natural language processing`, `adversarial ML`, `generative modelling`, `meta-learning`, `fairness`, `generalization`, `optimization`, `graphs`, `Bayesian methods`, and `Other`.[1] In total, 1605 papers from 2020 fell into the above categories, and 772 out of 1605 papers fell into multiple categories.

Figure 1: **Breakdown of papers and review outcomes by topic,** ICLR 2020

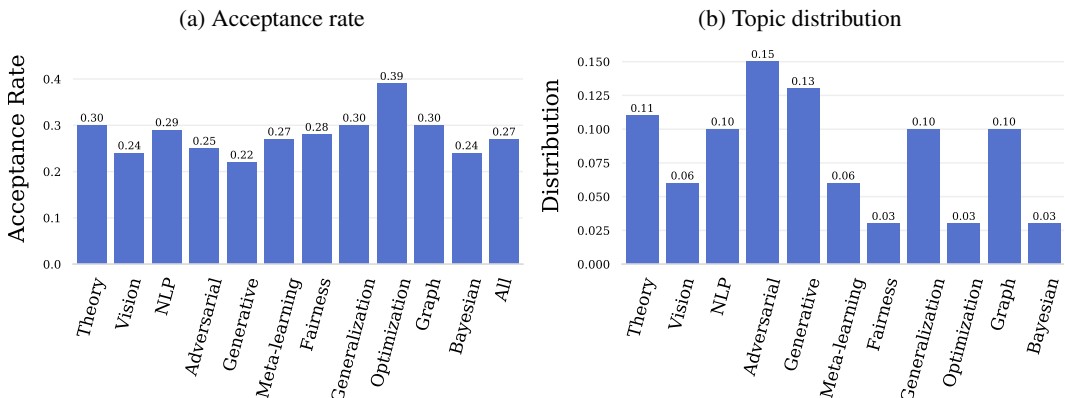

(a) Acceptance rate

(b) Topic distribution

---

[1]See Appendix A.1 for details on the keywords that we chose.

## 3 BENCHMARKING THE REVIEW PROCESS

We measure the quality of the review process by several metrics, including reproducibility and correlation between scores and paper impact. We also study how these key metrics have evolved over time, and provide suggestions on how to improve the review process.

### 3.1 REPRODUCIBILITY OF REVIEWS

The reproducibility of review outcomes has been a hot-button issue, and many researchers are concerned that reviews are highly random. This issue was brought to the forefront by the well-known "NIPS experiment" (Lawrence & Cortes, 2014), in which papers were reviewed by two different groups.[2] This experiment yielded the observation that 43% of papers accepted to the conference would be accepted again if the conference review process were repeated. In this section, we conduct an analysis to quantify the randomness in the review process for papers submitted to ICLR from 2017 to 2020. Our analysis is post hoc, unlike the controlled experiment from NeurIPS 2014.

*Monte-Carlo simulation* To produce an interpretable metric of reproducibilty that accounts for variation in both reviewers and area chairs, we use Monte-Carlo simulations to estimate the "NIPS experiment" metric: if an accepted paper were reviewed anew, would it be accepted a second time?

Our simulation samples from review scores from all 2560 papers submitted to ICLR in 2020. We simulate area chair randomness using a logistic regression model that predicts the AC's accept/reject decision as a function of a paper's mean reviewer score. This model is fit using 2020 paper data, and the goodness of fit of this model is explored in Appendix A.8.

To estimate reproducibility, we treat the empirical distribution of mean paper scores as if it were the true distribution of "endogenous" paper quality. We simulate a paper by (i) drawing a mean/endogenous score from this distribution. We then (ii) examine all 2020 papers with similar (within 1 unit) mean score, and compute the differences between the review scores and mean score for each such paper. We (iii) sample 3 of these differences at random, and add them to the endogenous score for the simulated paper to produce 3 simulated reviewer scores. We then (iv) use the simulated reviewer scores as inputs to the 2020 logistic regression model to predict whether the paper is accepted. Finally, we (v) generate a second set of reviewer scores using the same endogenous score, and use the logistic regression to see whether the paper is accepted a second time.

We run separate simulations for years 2017 through 2020 with their own respective logistic regression models. We observe a downward trend in reproducibility, with scores decreasing from 75% in 2017, to 70% in 2018 and 2019, to 66% in 2020.

Reproducibility scores by topic appear in Figure 10 of the Appendix. We observe that `Computer Vision` papers have the lowest reproducibility score with 59%, while `Generalization` papers have the highest score with 70%, followed by `Theory` papers with 68%.

*Analysis of Variance* A classical method for quantifying randomness in the review process is *analysis of variance* (ANOVA). In this very simple generative model, (i) each paper has an endogenous quality score that is sampled from a Gaussian distribution with learned mean/variance, and (ii) reviewer scores are sampled from a Gaussian distribution centered around the endogenous score, but with a learned variance. The ANOVA method generates unbiased estimates for the mean/variance parameters in the above model. An ANOVA using all of scores in 2020 finds that endogenous paper quality has a standard deviation of 2.95, while review scores have deviation 1.72. This can be interpreted to mean that the variation between reviewers is roughly 60% as large as the variation among papers. This does not account for the randomness of the area chair. Note the Gaussian model is a fairly rough approximation of the actual score distribution (ANOVA assumptions as discussed in Appendix A.2). Nonetheless, we report ANOVA results because this model is widely used and easily interpretable. Simulations in which papers and scores are drawn from an ANOVA model agree closely with simulations using empirical distributions (Appendix A.2).

---

[2]The NeurIPS conference was officially called NIPS prior to 2018.

## 3.2 SHOULD WE HAVE MORE REVIEWERS PER PAPER?

A number of recent conferences have attempted to mitigate the perceived randomness of accept/reject decisions by increasing the number of reviewers per paper. Most notably, efforts were made at NeuRIPS 2020 to ensure that most papers received 5 reviews. Unfortunately, simulations suggest that increasing the number of reviewers is not particularly effective at mitigating randomness.

We apply the above reproducibility model to review scores in ICLR 2020 while increasing the number of simulated reviews per paper, and we plot results in Figure 2. While reproducibility scores increase with more reviewers, gains are

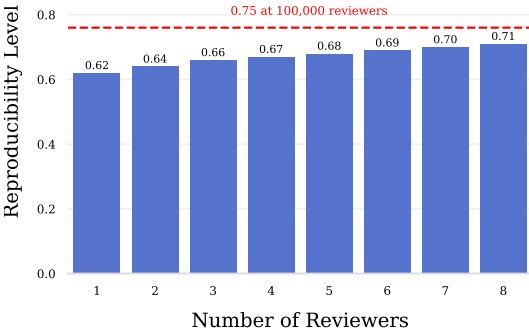

Figure 2: **Reproducibility by number of reviewers**, ICLR 2020.

marginal; increasing the number of reviewers from 2 to 5 boosts reproducibility by just 3%. Note that as the number of reviewers goes to infinity, the reproducibility score never exceeds 75%. This limiting behavior occurs because area chairs, who make final accept/reject decisions, vary in their standards for accepting papers. This uncertainty is captured by a logistic regression model in our simulation. As more reviewers are added, the level of disagreement among area chairs remains constant, while the standard error in mean scores falls slowly. The result is paltry gains in reproducibility. The reproducibility on rejected papers and on all papers can be found in Appendices A.5 and A.6, respectively.

Our study suggests that program chairs should avoid assigning large numbers of reviewers per paper as a tactic to improve the review process. Rather, it is likely most beneficial to use a small number of reviews per paper (which gives reviewers more time for each review), and add ad-hoc reviewers to selected papers in cases where first-round reviews are uninformative.

## 3.3 CORRELATION WITH IMPACT

Much of the conversation around the review process focuses on reproducibility and randomness. However, a perfectly reproducible process can still be quite bad if it reproducibly rejects high-impact and transformative papers. It is thought by some that the ML review process favors non-controversial papers with incremental theoretical results over papers with big new ideas.

In this section, we analyze whether reviewer scores correlate with paper impact. We measure impact using citation rate, calculated by dividing citation count by the number of days since

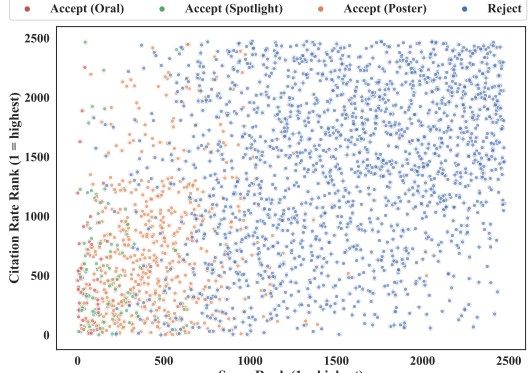

Figure 3: **Citation rank vs score rank for papers submitted to ICLR 2020**

the paper was first published online. After examining the data, citation rates are approximately linear in time although rare papers with extremely high citation count may exhibit exponential behavior. A similar study under the exponential model for citation count can be found in Appendix A.11. Though average scores for papers are roughly normally distributed, the distribution of citation rates is highly right-skewed and heavy tailed, with some papers receiving thousands of citations and others receiving none. To mitigate this issue, we use Spearman's rank correlation coefficient as a metric to measure the strength of the nonlinear relationship between average review score and citation rate.

We assign each paper a "citation rank" (the number of papers with lower citation rate) and "score rank" (the number of papers with lower mean reviewer score), then calculate Spearman's rank correlation coefficient between citation rank and score rank for all submitted papers. At first glance, the Spearman correlation ($0.46$, $p < 0.001$) seems to indicate a moderate relationship between scores and citation impact. However, we find that most of this trend is easily explained by the fact that higher scoring papers are accepted to the conference, and gain citations from the public exposure at

ICLR. This trend is clearly seen in Figure 3, in which there is a sharp divide between the citation rates of accepted and rejected papers.

To remove the effect of conference exposure, we calculate Spearman coefficients separately on accepted posters and rejected/retracted papers, obtaining Spearman correlations of 0.17 and 0.22, respectively. While these small correlations are still statistically significant ($p < 0.001$) due to the large number of papers being analyzed, the effects are quite small; an inspection of Figure 3 does not reveal any visible relationship between scores and citation rates within each paper category.

Spotlight papers had a Spearman correlation of -0.043 ($p = 0.66, n = 107$), and oral papers had Spearman correlation -0.0064 ($p = 0.97, n = 48$).

### 3.4 SHOULD WE DISCOURAGE RE-SUBMISSION OF PAPERS?

It is likely that papers that appeared on the web long before the ICLR deadline were previously rejected from another conference. We find that papers that have been online for more than 3 months at the time of the ICLR submission date, in fact, have *higher* acceptance rates at ICLR as well as higher average scores. This fact is observed in each of the past 4 years (see Figure 4a). As these papers are likely to have been previously rejected, our findings suggest that resubmission should not be discouraged, as these papers are more likely to be accepted than fresh submissions. Interestingly, papers which did not appear online prior to the submission deadline, denoted by "not visible online", have the lowest acceptance rate.

Figure 4: **Papers statistics by time online before submission deadline**

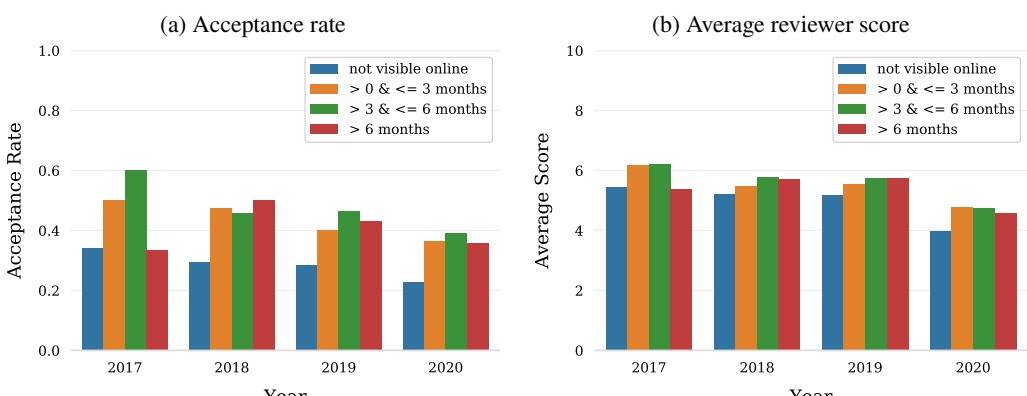

(a) Acceptance rate          (b) Average reviewer score

### 4 HAS THE REVIEW PROCESS GOTTEN "WORSE" OVER TIME?

Our dataset shows that reproducibility scores, correlations with impact, and reviewer agreement have all gone down over the years. Downward temporal trends were already seen in reproducibility scores. Spearman correlations between scores and citations decreased from 0.582 in 2017 to 0.471 in 2020. We also measure the "inter-rater reliability," which is a well-studied statistical measure of the agreement between reviewers (Hallgren, 2012), using the Krippendorf alpha statistic (Krippendorff, 2011; De Swert, 2012). We observe that alpha decreases from 56% in 2017 to 39% in 2020. More details can be found in Appendix A.3.

It should be noted that ICLR changed the rating scale used by reviewers in 2020 and also decreased the paper acceptance rate. These factors may be responsible in part for the sharp fall-off in reproducibility in 2020. We address this issue by bumping 2020 reviewer scores up so that the acceptance rate matches 2019, and then re-running our simulations. After controlling for acceptance rate in this way, we still observe an almost identical falloff in 2020, indicating that the change in reproducibility is not explained by the decreasing acceptance rate alone.

### 5 BIASES

We explore sources of bias in the review process, including institutional bias and bias toward reputable authors. We also discuss how gender correlates with outcomes.

## 5.1 INSTITUTIONAL BIAS

Does ICLR give certain institutions preferential treatment? In this section, we see whether the rank of institution (as measured by *CS Rankings*) influences decisions. We focus on academic institutions because they fall within a widely known quantitative ranking system. We found that 85% of papers across all years (87% in 2020) had at least one academic author. In the event that a paper has authors from multiple institutions, we replicate the paper and assign a copy of it to each institution. Figure 13 in Appendix A.7 plots the average of all paper scores received by each institution, ordered by institution rank. We find no meaningful trend between institution rank and paper scores.

*Area Chair Bias*   It is difficult to detect institutional bias in reviews because higher scores among more prestigious institutions could be explained by either bias or differences in paper quality. It is easier to study institutional bias at the AC level, as we can analyze the accept/reject decisions an AC makes while controlling for paper scores (which serves as a proxy for paper quality).

To study institutional bias at the AC level, we use a logistic regression model to predict paper acceptance as a function of (i) the average reviewer score, and (ii) an indicator variable for whether a paper came from one of the top 10 most highly ranked institutions.

We found that, even after controlling for reviewer scores, being a top ten institution leads to a boost in the likelihood of getting accepted. This boost is equivalent to a 0.15 point increase in the average reviewer score ($p = 0.028$).[3] Regression details can be found in Table 4 located in Appendix A.9.

Prestigious institutions are more likely than others to have famous researchers, and the acceptance bias could be attributed to the reputation of the researcher rather than the university. In Section 5.3 we control for last-author reputation and still find a significant effect for institution rank (Table 1).

We find that most of the institutional bias effect is explained by 3 universities: Carnegie Mellon, MIT, and Cornell. We repeat the same study, but instead of a top 10 indicator, we use a separate indicator for each top 10 school. Most institutions have large p-values, but CMU, MIT, and Cornell each have statistically significant boosts equating to 0.31, 0.31, and 0.58 points in the average reviewer score, respectively. See Tables 5 and 6 in Appendix A.9.

While 2020 area chairs came from a diverse range of universities, 26/115 of them had a Google or Deepmind affiliation, which lead us to investigate bias effects for large companies (Table 7, A.9). The 303 papers from Google/Deepmind have an insignificant boost (0.007 points, $p = 0.932$) as did the 110 papers from Facebook (0.07 points, $p = 0.630$). However, the 92 papers from Microsoft had a strong penalty (-0.50 points, $p = 0.003$).

## 5.2 THE ARXIV EFFECT

Is it better to post a paper on arXiv before the review process begins, or to stay anonymous? To explore the effects of arXiv, we introduce a new indicator variable that detects if a paper was posted online by a date one week after the ICLR submission deadline. We fit logistic regression models to predict paper acceptance as a function of (i) the average score given by the reviewers of a paper, and (ii) an indicator variable for papers that were visible on arXiv during the review process.

We fit models separately on 3 subsets of the 2020 papers: schools not in the top 10 (Table 9, A.9), top 10 schools excluding CMU/MIT/Cornell (Table 11, A.9), and papers from CMU, MIT, and Cornell (Table 10, A.9). We find that by having papers visible on arXiv during the review process, CMU, MIT, and Cornell as a group receive a boost of 0.67 points ($p = 0.001$). Conversely, institutions not in the top 10 received a statistically insignificant boost (0.08 points, $p = 0.235$), as did top 10 schools excluding CMU, MIT, and Cornell (0.05 points, $p = 0.737$).

Overall, we found that papers appearing on arXiv tended to do better (after controlling for reviewer scores) across the board, including schools that did not make the top 50, in addition to Google, Facebook, and Microsoft. However, these latter effects are not statistically significant.

---

[3]We compute this effect by taking the ratio of the indicator coefficient and the average reviewer score coefficient, which represents the relative impact on the likelihood of a paper being accepted.

### 5.3 REPUTATION BIAS

Are papers by famous authors more likely to be accepted even after controlling for reviewer scores? We quantify the reputation of an author by computing `log(citations/papers+1)` for each author, where `citations` is the total citations for the author, and `papers` is their total number of papers (as indexed by Semantic Scholar). The raw ratio `citations/papers` has a highly skewed and heavy tailed distribution, and we find that the log transform of the ratio has a symmetric and nearly Gaussian distribution for last authors (see Figure 15, Appendix A.8).

We run a logistic regression predicting paper acceptance as a function of (i) the average reviewer score, (ii) the reputation index for the last author, and (iii) an indicator variable for whether a paper came from a top-ten institution. We include the third variable to validate the source of bias, because institution and author prestige may be correlated. Table 1 shows that last author reputation has a statistically significant positive relationship with AC decisions when controlling mean reviewer score and institution. Taking into account the distribution of the last author reputation index (Figure 15, Appendix A.8), these coefficients indicate that a last author near the mean of the author reputation index will gain a boost equivalent to a 0.16 point increase in average reviewer score, while last authors at the top (2 standard deviations above the mean) receive a boost equivalent to a 0.29 point increase.

Table 1: **Relationship between author reputation/institution and acceptance.** This logistic regression model has 93% accuracy on a hold-out set containing 30% of 2020 papers.

| Variable | Coefficient | Std. Error | Z-score | p-value |
|---|---|---|---|---|
| mean reviewer score | 2.4354 | 0.098 | 24.807 | 0.000 |
| log(citations/papers+1) | 0.1302 | 0.057 | 2.270 | 0.023 |
| top ten school? | 0.2969 | 0.165 | 1.802 | 0.072 |
| constant | -13.6620 | 0.562 | -24.313 | 0.000 |

### 5.4 GENDER BIAS

There is a well-known achievement gap between women and men in the sciences. Women in engineering disciplines tend to have their papers accepted to high impact venues, and yet receive fewer citations (Ghiasi et al., 2015; Larivière et al., 2013; Lariviere, 2014; Rossiter, 1993). These citation disparities exist among senior researchers at ICLR. Over their whole career, male last authors from 2020 have on average 44 citations per publication while women have 33. This gap in total citations can be attributed in part to differences in author seniority; the 2019 Taulbee survey reports that women are more severely under-represented among senior faculty (15.7%) than among junior faculty (23.9%) (Zweben & Bizot, 2020), and more senior faculty have had more time to accumulate citations. To remove the effect of time, we focus in on only 2020 ICLR papers, where male last authors received on average 4.73 citations to date compared to 4.16 citations among women. This disparity flips for first authors; male first authors received just 4.4 citations vs 6.2 among women.

We find that women are under-represented at ICLR relative to their representation in computer science as a whole. In 2019, women made up 23.2% of all computer science PhD students (Zweben & Bizot, 2020) in the US. However, only 12.1% of publications at ICLR 2020 from US universities have a female first author. While women make up 22.6% of CS faculty in the US, they make up only 12.8% of last authors from US institutions at ICLR 2020. Similar trends occur in the experimental sciences (Gaule & Piacentini, 2018; Pezzoni et al., 2016; Kato & Song, 2018). Among papers from top-10 schools in the US, women make up 12.3% of first authors and 13.4% of last authors.

We observe a gender gap in the review process, with women first authors achieving a lower acceptance rate than men (23.5% vs 27.1%). [4] This performance gap is reflected in individual reviewer scores, as depicted by histograms in Figure 5. Women on average score 0.16 points lower than their male peers. A Mann-Whitney U test generated a (two sided) p-value of 0.085.[5] No significant differences were observed when scores were split by the gender of last author.

---

[4]A large gap in acceptance rates is not seen for last authors. See Table 13 in Appendix A.10.

[5]We use a non-parametric test due to the non-Gaussian and sometimes multi-modal distribution of paper scores.

Figure 5: **Histogram of average scores by gender, first and last author**, ICLR 2020.

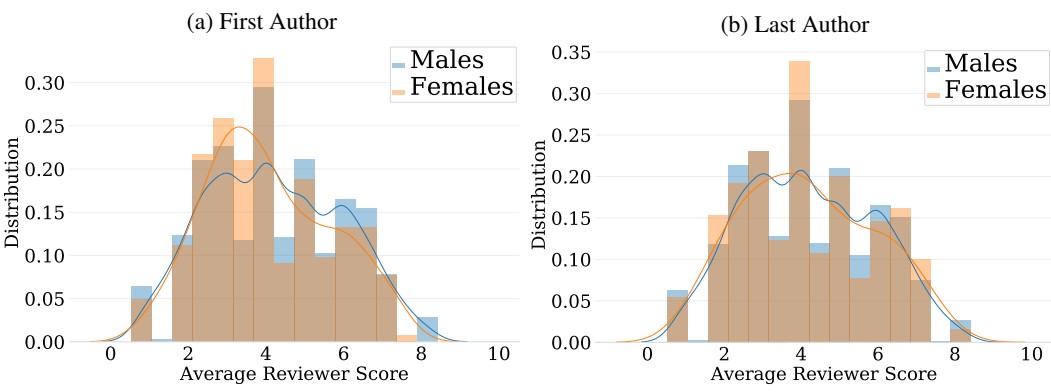

As noted in Section 1, there is a wide range of acceptance rates among the different research topics at ICLR. Among papers that fall into a labeled topic (excluding `other`), male and female first author papers are accepted with rates 27.6% and 22.1%, respectively. One may suspect that this disparity is explained by differences in the research topic distribution of men and women, as shown in Figure 6b. However, we find that differences in topic distribution are actually more favorable to women than men. If women kept their same topic distribution, but had the same per-topic acceptance rates as their male colleagues, we should expect women to achieve a rate of 27.8% among topic papers (slightly more than their male colleagues) – a number far greater than the 22.1% they actually achieve.

Figure 6: **Acceptance rate and Distribution by topic for male and female first authors**

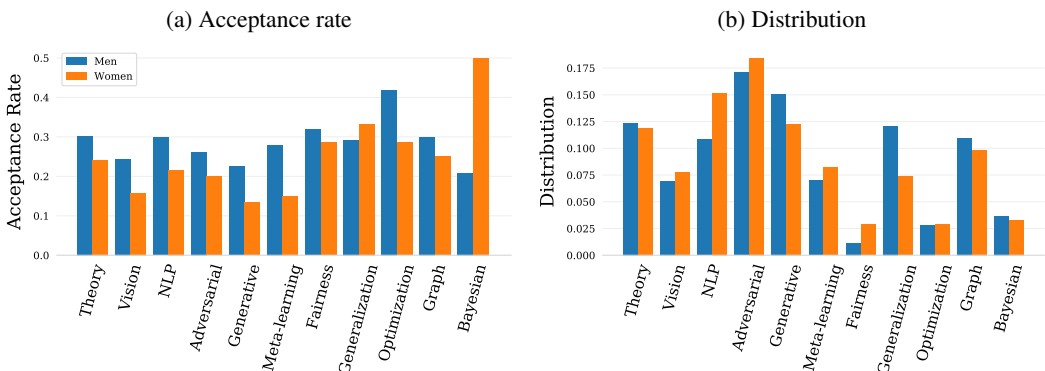

We test for area chair bias with a regression predicting paper acceptance as a function of review scores and gender. We did not find evidence for AC bias (Figure 12, Appendix A.9).

## 6 CONCLUSION

We find the level of reproducibility at ICLR (66% in 2020) to be higher than we expect when considering the much lower acceptance rate (26.5%), which seemingly contradicts the notion that reviews are "random." Nonetheless, many authors find large swings in reviews as they resubmit papers to different conferences and find it difficult to identify a home venue where their ideas feel respected. We speculate that the perceived randomness of conference reviews is the result of several factors. First, differences in paper matching and bidding systems used by different conferences can sway the population of reviewers that are recommended to bid on an article, resulting in a major source of inter-conference randomness that is not represented in the above intra-conference study. Second, the influx of researchers from disparate backgrounds means that the value system of a paper's reviewers is often mismatched with that of its authors.

Finally, we interpret the empirical studies in this paper to suggest that acceptance rates at mainstream conferences are just too low. This is indicated by the prevalence of paper re-submission within the ML community, combined with the relatively high marks of re-submitted papers we observed here. When the number of papers with merit is greater than the number of papers that will be accepted, it

is inevitable that decisions become highly subjective and that biases towards certain kinds of papers, topics, institutions, and authors become endemic.

In our work, we study a number of hand-selected factors for which sufficient data exists to draw statistically significant conclusions. Additional interesting factors and confounding variables likely exist. However, multi-factor studies are limited by the number of historic ICLR submissions. For example, it is not possible to simultaneously study the effect of gender and institution rank because there are not nearly enough women authors at ICLR to get statistically meaningful results regarding gender after the dataset is sliced into bins based on institution. Despite the limitations stemming from the high number of interesting variables along with limited data, this study yields a number of findings which are informative for the machine learning community and conference organizers.

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

## A  APPENDIX

Additional experimental details and results are included below.

### A.1  BREAKDOWN OF PAPERS BY TOPIC

We identify a paper with a topic if it contains at least one of the relevant keywords. The topics [and keywords] used were `theory` [theorem, prove, proof, theory, bound], `computer vision` [computer vision, object detection, segmentation, pose estimation, optical character recognition, structure from motion, facial recognition, face recognition], `natural language processing` [natural language processing, nlp, named-entity, machine translation, language model, word embeddings, part-of-speech, natural language], `adversarial ML` [adversarial attack, poison, backdoor, adversarial example, adversarially roburst, adversarial training, certified roburst, certifiably roburst], `generative modelling` [generative adversarial network, gan, vae, variational autoencoder], `meta-learning` [few-shot, meta learning, transfer learning, zero-shot], `fairness` [gender, racial, racist, biased, unfair, demographic, ethnic], `generalization` [generalization], `optimization` [optimization theory, convergence rate, convex optimization, rate of convergence, global convergence, local convergence, stationary point], `graphs` [graph], `Bayesian methods` [bayesian], and `Other` which includes the papers that do not fall into any of the above categories.

## A.2 ANOVA METHOD

Here we show histograms of mean scores over all papers, and we plot the variance of individual reviewer scores as a function of paper quality. We observe that the distribution of paper scores has overall Gaussian structure, however this structure is impacted by the fact that certain mean scores can be achieved using a large number of individual score combinations, while others cannot, resulting in some spiked behavior. We also find at the homogeneity assumptions of ANOVA (i.e., the reviewer variance remains the same for all papers) is not well satisfied for all conference years. For this reason, we do not want to rely heavily on ANOVA for our reliability analysis, and we were motivated to construct the Monte Carlo simulation model. Interestingly, repeatability simulations using our ANOVA model yield results in close agreement with the Monte Carlo model.

Figure 7: **Histogram for average scores of all papers**. This plot demonstrates that the average scores in each year follows a roughly Gaussian distribution. In some years there are particular scores that appear to be chosen relatively infrequently, causing multi-modal behavior. This is what inspired up to use a Monte-Carlo simulation in addition to the ANOVA results. Interestingly, our Monte-Carlo simulations produce similar results regardless of whether we sample the empirical score distribution, or the Gaussian approximation from ANOVA.

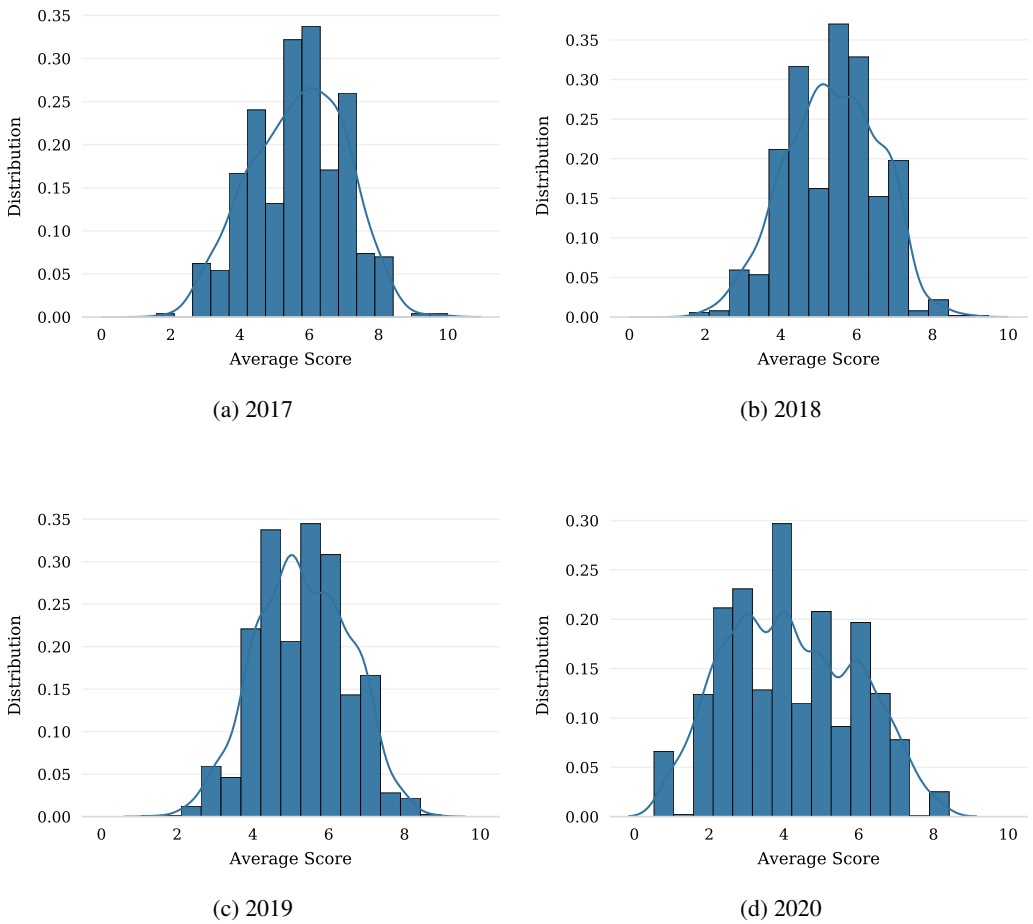

(a) 2017

(b) 2018

(c) 2019

(d) 2020

Figure 8: **Bar plot for average standard deviation within each score range**. This plot demonstrates that the standard deviations are roughly equal across different ranges in 2017, 2018 and 2019. However, we observe that the standard deviations from score ranges between 3 and 6 are higher than the rest in 2020. Note that in 2018 and 2019, there are no papers that have an average score greater than 9.

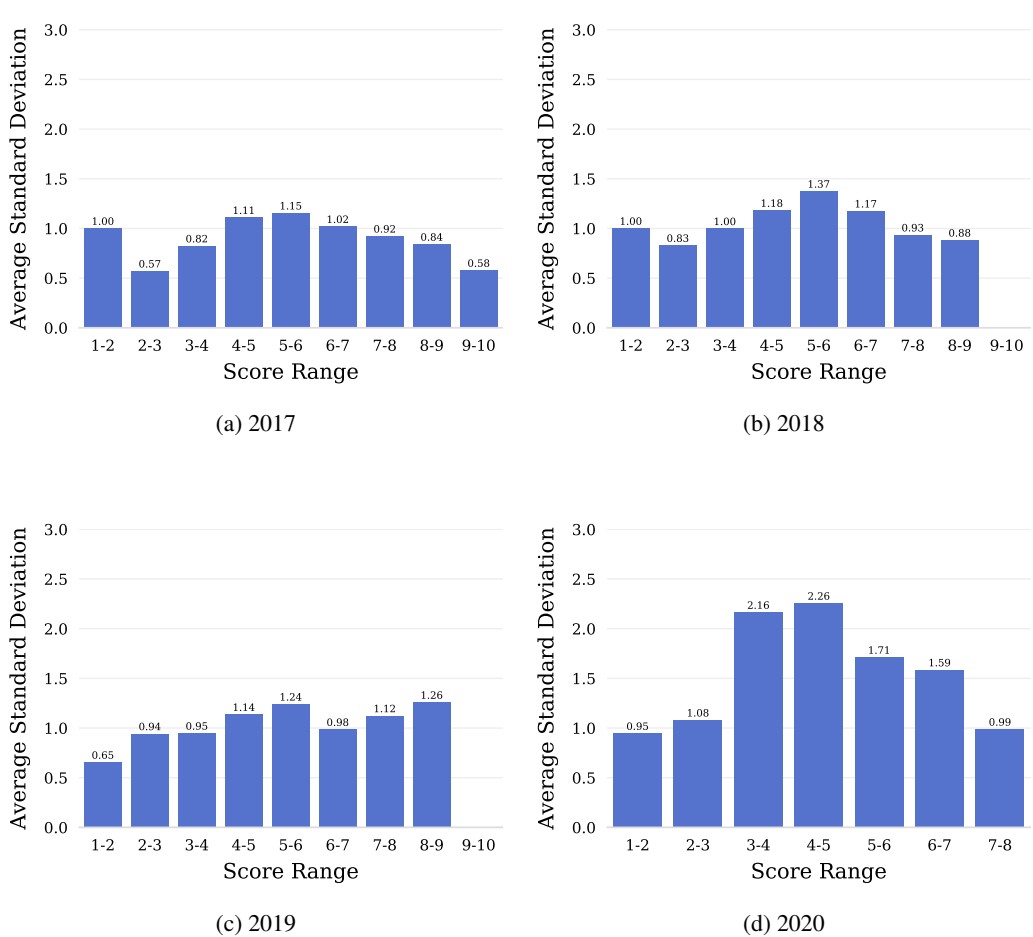

(a) 2017

(b) 2018

(c) 2019

(d) 2020

Table 2: **Standard deviations from ANOVA model.**. Summary for standard deviation between all average scores and standard deviation for review scores within a paper.

| Year | Std. Between | Std. Within |
|------|--------------|-------------|
| 2017 | 2.3334 | 1.0307 |
| 2018 | 2.1080 | 1.1947 |
| 2019 | 2.1038 | 1.1036 |
| 2020 | 2.9595 | 1.7209 |

Figure 9: **Reproducibility score over time**. This plot compares the reproducibility score over time using the ANOVA model.

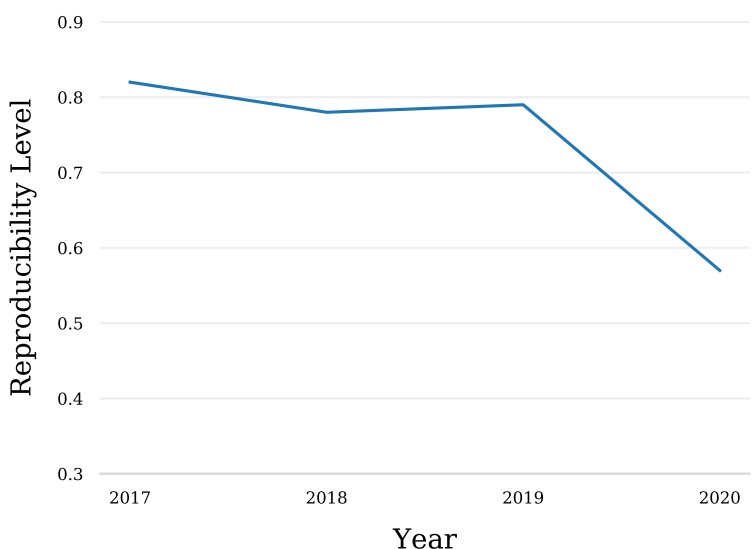

### A.3 KRIPPENDORF ALPHA STATISTIC

Krippendorff's alpha is a general statistical measure of agreement among raters, observers, or coders of data, designed to indicate their reliability. The advantage of this statistic is that it supports ordinal ratings, handles missing data, and is applicable to any number of raters, each assigning ratings to any other unit of analysis. As a result, the Krippendorff's alpha is an appropriate choice for our dataset given that there are many reviewers in the conference, each reviewer in the conference usually rates several papers, and no reviewer rates all papers. Since the computation of the Krippendorf's alpha depends only on the frequency of scores given to each paper, we can calculate the statistics despite the anonymous nature of the review process. We observe a downward trend in agreement between reviewers, with alpha of 56% in 2017, 42% in 2018, 47% in 2019, and 39% in 2020 where 100% indicates perfect agreement and 0% indicates absence of agreement.

## A.4 REPRODUCIBILITY LEVEL BY TOPIC

Figure 10: **Reproducibility level by topic**, ICLR 2020.

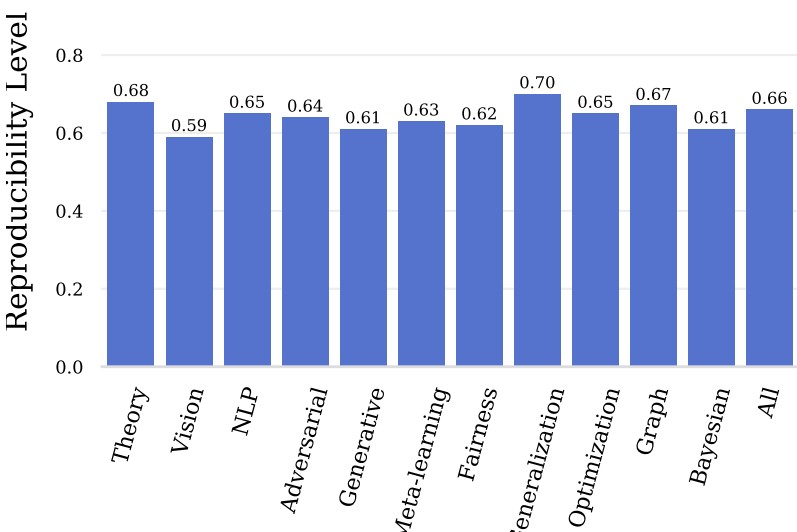

## A.5 REPRODUCIBILITY LEVEL BY NUMBER OF REVIEWERS FOR REJECTED PAPERS

Figure 11: **Reproducibility level by number of reviewers**, ICLR 2020. The reproducibility level denotes the probability that a rejected paper is rejected a second time.

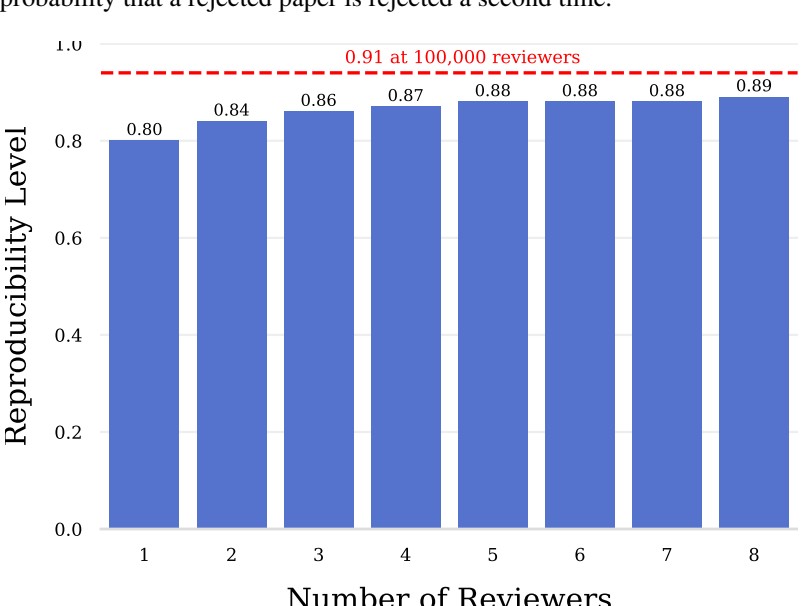

## A.6 REPRODUCIBILITY LEVEL BY NUMBER OF REVIEWERS FOR ALL PAPERS

Figure 12: **Reproducibility level by number of reviewers**, ICLR 2020. The reproducibility level denotes the probability that a paper achieves the same outcome twice.

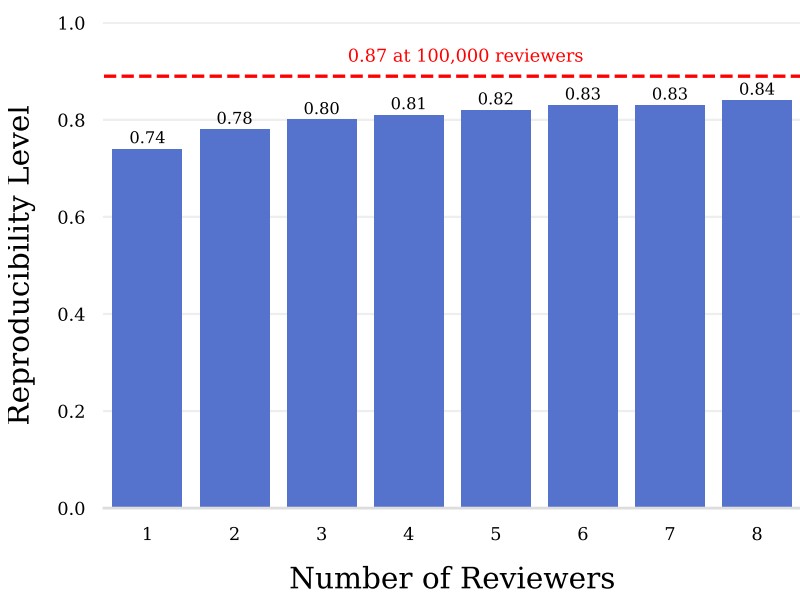

## A.7 CORRELATIONS BETWEEN INSTITUTION RANK AND PAPER SCORES

Figure 13: **Institution rank vs average score for the top 100 institutions.** For each institution and each paper decision type, all reviewer scores were averaged to generate a data point. Shaded areas represent 95% confidence intervals.

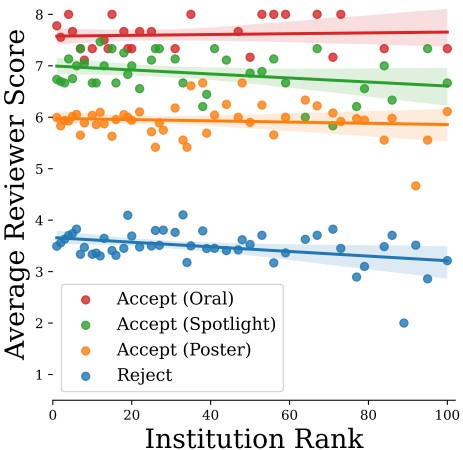

A.8    VERIFYING ASSUMPTIONS OF STATISTICAL TESTS

Figure 14: **Logistic Regression Assumptions Checking**. We use a Hosmer-Lemeshow plot to check the assumption for our logistic regression model. The null hypothesis is that there is no difference between observed deciles and those predicted by the logistic model. If the values are the same, we have a model that fits perfectly. This graph compares the observed number of accepted papers and expected number of accepted papers across multiple groups.

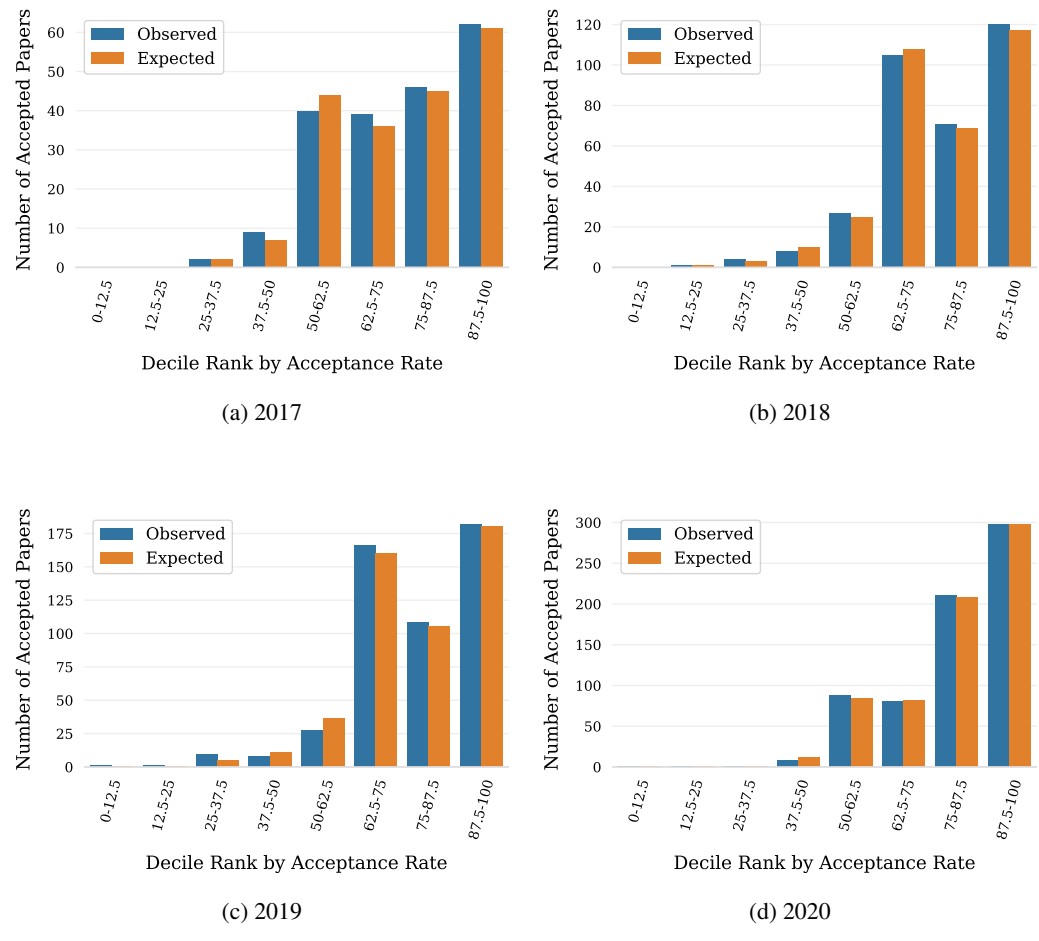

Table 3: **Stukel's Test.** We also use Stukel's test, another goodness-of-fit test for logistic regression. A statistically significant p-value indicates a poor fit. This test validates the logistic regression models on years 2017, 2018, and 2020, while it indicates that the 2019 model provides a poorer fit.

| Year | p-value |
|------|---------|
| 2017 | 0.46 |
| 2018 | 0.37 |
| 2019 | 0.00 |
| 2020 | 0.75 |

Figure 15: **Histogram for last author reputation index**. This plot demonstrates that the transformation `log(citations/publication + 1)` achieves a roughly Gaussian distribution with a mean of 2.95 and a standard deviation of 1.28. Last authors were taken from the 2020 conference only.

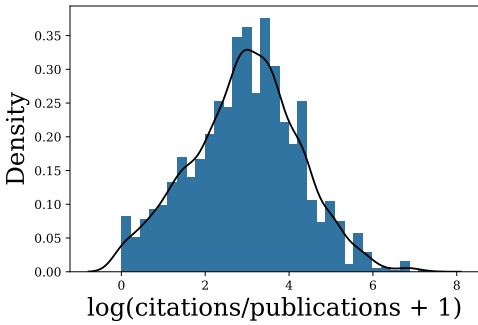

## A.9 ADDITIONAL LOGISTIC REGRESSION SUMMARIES

Table 4: **Top 10 institutions' impact on acceptance, together**, ICLR 2020. Summary of the Logistic Regression predicting paper acceptance as a function of mean reviewer score and a top 10 indicator. The accuracy of the classifier was 92% on a hold-out set containing 30% of 2020 papers

| Variable | Coefficient | Std. Error | Z-score | p-value |
|---|---|---|---|---|
| mean reviewer score | 2.4350 | 0.097 | 25.118 | 0.000 |
| top ten school? | 0.3536 | 0.161 | 2.194 | 0.028 |
| constant | -13.2707 | 0.526 | -25.212 | 0.000 |

Table 5: **Top 10 institutions' impact on acceptance, separate**, ICLR 2020. Logistic regression summary for predicting paper acceptance. The top 10 institution ranks each have their own indicator variable. Statistically significant effects are in bold. The accuracy of the classifier was 92% on a hold-out set containing 30% of 2020 papers

| Variable | Coefficient | Std. Error | Z-score | p-value |
|---|---|---|---|---|
| mean reviewer score | 2.4600 | 0.099 | 24.949 | 0.000 |
| **Carnegie Mellon** | **0.7510** | 0.363 | 2.071 | **0.038** |
| **MIT** | **0.7498** | 0.357 | 2.100 | **0.036** |
| U. Illinois, Urbana-Champaign | 0.6698 | 0.535 | 1.252 | 0.211 |
| Stanford | -0.0312 | 0.371 | -0.084 | 0.933 |
| U.C. Berkeley | 0.2299 | 0.337 | 0.681 | 0.496 |
| U. Washington | -0.8189 | 0.684 | -1.198 | 0.231 |
| **Cornell** | **1.4267** | 0.607 | 2.350 | **0.019** |
| Tsinghua U. & U. Michigan (tied) | -0.0118 | 0.369 | -0.032 | 0.974 |
| ETH Zurich | 0.0608 | 0.640 | 0.095 | 0.924 |
| constant | -13.4048 | 0.535 | -25.038 | 0.000 |

Table 6: **Institution & last author reputation impact on acceptance, CMU, MIT, and Cornell**, ICLR 2020. Logistic regression predicting paper acceptance as a function of mean reviewer score, a last author reputation index, and 3 institution indicators. The accuracy of the classifier was 90% on a hold-out set containing 30% of 2020 papers.

| Variable | Coefficient | Std. Error | Z-score | p-value |
|---|---|---|---|---|
| mean reviewer score | 2.4511 | 0.099 | 24.718 | 0.000 |
| log(citations/paper + 1) | 0.1306 | 0.057 | 2.285 | 0.022 |
| Carnegie Mellon (rank 1) | 0.6730 | 0.368 | 1.829 | 0.067 |
| MIT (rank 2) | 0.6783 | 0.358 | 1.895 | 0.058 |
| Cornell (rank 7) | 1.3482 | 0.611 | 2.207 | 0.027 |
| constant | -13.7425 | 0.567 | -24.238 | 0.000 |

Table 7: **Institution impact on acceptance, Google, Facebook, and Microsoft**, ICLR 2020. Logistic regression predicting paper acceptance as a function of mean reviewer score and 3 institution indicators. The accuracy of the classifier was 91% on a hold-out set containing 30% of 2020 papers.

| Variable | Coefficient | Std. Error | Z-score | p-value |
|---|---|---|---|---|
| mean reviewer score | 2.3515 | 0.075 | 31.557 | 0.000 |
| google | -0.0177 | 0.209 | -0.085 | 0.932 |
| facebook | 0.1745 | 0.362 | 0.482 | 0.630 |
| microsoft | -1.1837 | 0.402 | -2.948 | 0.003 |
| constant | -12.7907 | 0.402 | -31.843 | 0.000 |

Table 8: **Visibility on arXiv during the review period, top 10 institutions** at ICLR 2020. Logistic regression predicting paper acceptance as a function of mean reviewer score and an indicator demarcating if the paper was on arXiv up to one week after the submission date. The accuracy of the classifier was 93% on a hold-out set containing 30% of 2020 papers.

| Variable | Coefficient | Std. Error | Z-score | p-value |
|---|---|---|---|---|
| mean reviewer score | 2.3904 | 0.190 | 12.551 | 0.000 |
| visible on arXiv? | 0.6599 | 0.288 | 2.294 | 0.022 |
| constant | -12.9311 | 1.014 | -12.754 | 0.000 |

Table 9: **Visibility on arXiv during the review period, institutions not in the top 10** at ICLR 2020. Logistic regression predicting paper acceptance as a function of mean reviewer score and an indicator demarcating if the paper was on arXiv up to one week after the submission date. The accuracy of the classifier was 92% on a hold-out set containing 30% of 2020 papers.

| Variable | Coefficient | Std. Error | Z-score | p-value |
|---|---|---|---|---|
| mean reviewer score | 2.4309 | 0.112 | 21.650 | 0.000 |
| visible on arXiv? | 0.2033 | 0.171 | 1.187 | 0.235 |
| constant | -13.3262 | 0.610 | -21.859 | 0.000 |

Table 10: **Visibility on arXiv during the review period, CMU, MIT, and Cornell** at ICLR 2020. Logistic regression predicting paper acceptance as a function of mean reviewer score and an indicator demarcating if the paper was on arXiv up to one week after the submission date. The accuracy of the classifier was 94% on a hold-out set containing 30% of 2020 papers.

| Variable | Coefficient | Std. Error | Z-score | p-value |
|---|---|---|---|---|
| mean reviewer score | 2.7144 | 0.372 | 7.295 | 0.000 |
| visible on arXiv? | 1.8285 | 0.551 | 3.317 | 0.001 |
| constant | -14.4988 | 1.967 | -7.372 | 0.000 |

Table 11: **Visibility on arXiv during the review period, top 10 institutions excluding CMU, MIT, and Cornell** at ICLR 2020. Logistic regression predicting paper acceptance as a function of mean reviewer score and an indicator demarcating if the paper was on arXiv up to one week after the submission date. The accuracy of the classifier was 91% on a hold-out set containing 30% of 2020 papers.

| Variable | Coefficient | Std. Error | Z-score | p-value |
|---|---|---|---|---|
| mean reviewer score | 2.3874 | 0.239 | 9.987 | 0.000 |
| visible on arXiv? | 0.1203 | 0.358 | 0.336 | 0.737 |
| constant | -12.9840 | 1.276 | -10.175 | 0.000 |

Table 12: **Gender impact on acceptance**, ICLR 2020. Logistic regression predicting paper acceptance as a function of mean paper reviewer score and a gender indicator variable for both the first and last author. The accuracy of the classifier was 93% on a hold-out set containing 30% of 2020 papers. The results were inconclusive. Papers with last authors that were unlabelled were excluded.

| Variable | Coefficient | Std. Error | Z-score | p-value |
|---|---|---|---|---|
| mean reviewer score | 2.3956 | 0.109 | 21.877 | 0.000 |
| gender indicator, first author | -0.1045 | 0.266 | -0.392 | **0.695** |
| constant | -13.0241 | 0.633 | -20.590 | 0.000 |
| | | | | |
| mean reviewer score | 2.3813 | 0.107 | 22.349 | 0.000 |
| gender indicator, last author | -0.0520 | 0.272 | 0.191 | **0.849** |
| constant | -13.0441 | 0.627 | -20.800 | 0.000 |

A.10  ADDITIONAL STATISTICS ON GENDER

Table 13: **Summary of gender statistics.** Some percents do not add up to 100%; we were unable to label 5.6% of first authors and 2.5% of last authors.

| | First Author | | Last Author | |
|---|---|---|---|---|
| Statistic | *Male* | *Female* | *Male* | *Female* |
| Percent of Papers | 83.75% | 10.63% | 87.81% | 9.65% |
| Percent of Accepted Papers | 84.43% | 9.32% | 88.50% | 9.46% |
| Acceptance Rate | 27.05% | 23.53% | 27.05% | 26.32% |
| Average Reviewer Score | 4.21 | 4.06 | 4.20 | 4.18 |

### A.11  CORRELATION WITH IMPACT UNDER AN EXPONENTIAL MODEL FOR THE GROWTH RATE OF CITATION COUNT

Here, we calculate citation rate as $\log(\text{citations} + 1)/(\text{time})$. As with our experiments under the linear model in the main body, Figure 16 shows no visible relationship between scores and citation rates within each paper category. We obtain a Spearman correlation of 0.46 ($p < 0.001$) for all papers. Accepted posters and rejected papers have Spearman correlations of 0.13 ($p = 0.002$) and 0.21 ($p < 0.001$) respectively. Spotlight papers have Spearman correlation 0.045 ($p = 0.65, n = 107$) and oral papers have Spearman correlation 0.13 ($p = 0.39, n = 48$). Thus, we observe the same behavior under this model; specifically, score only has a small relationship with impact within decision categories.

Figure 16: **Log-transformed citation rate rank vs score rank for papers submitted to ICLR 2020**.

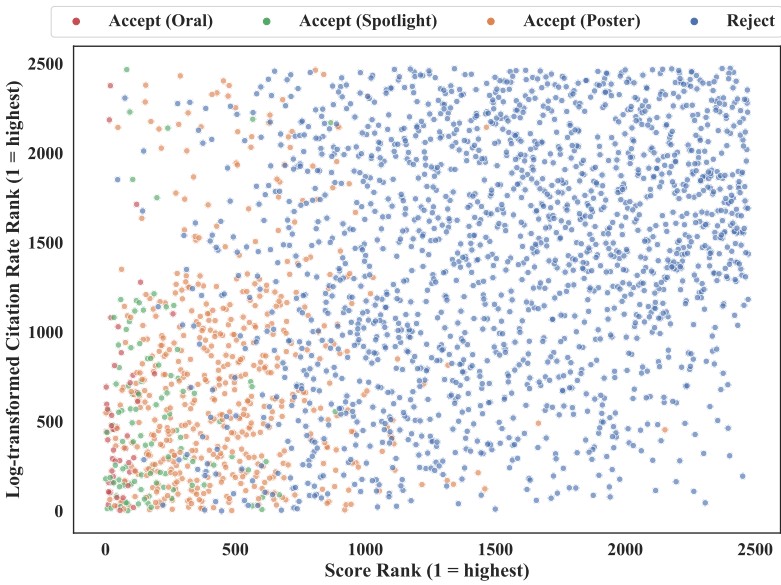

