# OpenReview forum: "An Open Review of OpenReview: A Critical Analysis of the Machine Learning Conference Review Process"
_ICLR.cc/2021/Conference — Reject_

### Official Review · AnonReviewer2 · 2020-10-25
**An empirical study on conference reviewing process**

**Rating:** 6
**Confidence:** 3

**Review:**


In this paper, the authors conduct an empirical study on major machine learning reviewing process. The goal of the study is to answer questions like how is the reproducibility or what is the bias in the review process.

In general the paper makes some timely attempts for the conference review process. Here are some of my concerns:

- In Sec 3.1, the authors basically assume that papers with similar mean review scores have similar variability in scores. I think more justifications are needed for this assumption.

- In Sec 5.1, the authors only consider the average reviewer score, and an indicator variable for whether a paper came from one of the top 10 most highly ranked institutions as two factors for ACs to make the decision. I think the ACs decision process is too simplified. The same problem exists for other sections when the authors try to fit a logistic regression model between the acceptance decisions and a number of factors that the authors pre-select. The authors may want to clearly explain if only these factors contribute to the final decision (other confounding issues matter?), or if the relationship could be captured by a LR model.

- In Sec 5.4, the authors assigned labels based on gendered pronouns appearing on personal webpages when possible, and on the use of canonically gendered names otherwise. The authors should report some statistics for how accurate this process is.

---

> ### Author Response · Authors · 2020-11-24
> **Thank you for your feedback.**
>
> Thank you for your feedback.  We address your comments below.
>
> 1) In the appendix, we examine the statistical assumptions of ANOVA, and we included a plot of variance as a function of reviewer score in order to justify this choice. We found that variance was similar for papers with moderate and borderline scores, and variance was somewhat lower for papers with extremely high or low marks.  This latter category of papers is uncommon.
>
> 2) It is true that our study focuses on a number of hand chosen factors.   We have made efforts to sort out several confounding variables, and in fact a number of studies that appear in the paper were the result of us trying to remove a confounding variable (e.g. Appearance on arxiv, breakdown of papers over topics, and the removing the effect of paper acceptance on citation rates).   We would also point out that our ability to do multi-factor studies is limited by the size of our dataset.   For example it is not possible to simultaneously study the effect of gender and institution rank because there are not nearly enough women authors at ICLR to get statistically meaningful results regarding gender after the dataset is sliced into bins based on institution.   Finally, there are certainly many potential confounding variables that we did not eliminate, and many that we are not aware of.  We cannot control for all of these confounders, but we think it’s important to acknowledge that they exist. We will add an acknowledgment of this weakness in the conclusion.
>
> 3) We used a two-step process to label gender.  First, we do a manual search for an author’s webpage or CV, and use the gender assignment of their self-selected pronouns when possible.  If this information is not available, then we assign a gender label based on the author’s name.  Note that name-based gender labelling is commonly done in the gender studies literature, and studies have shown that name-based gender labelling is reasonably accurate in many scenarios, with over 98% accuracy observed using a range of different name databases when the name labeling system is allowed to abstain if presented with a gender-ambiguous name (see [1] and [2] below).  We chose our two-step process over a totally automated name/gender database lookup process for two reasons.  First, it is known that name-gender databases tend to be largely built using western data sources and tend to have lower accuracy on names of Asian origin [1].  Furthermore, these databases may fail due to cultural differences in how names are gendered across regions (e.g., “Andrea” is a canonically female name in the US, but canonically male in Italy).  Because of the highly international nature of ICLR (and in particular the large number of authors from South and East Asia), we wanted a closely supervised process with humans in the loop rather than relying solely on a database.  Second, it was our hope to identify a large enough number of authors using gender-neutral pronouns to say something about the representation and performance of this group at ICLR. Unfortunately, we did not identify a large enough cohort of authors choosing gender-neutral pronouns to make statements that would have statistical significance and respect author privacy, and so we chose not to address this issue.
>
> [1] Santamaría, Lucía, and Helena Mihaljević. "Comparison and benchmark of name-to-gender inference services." PeerJ Computer Science 4 (2018): e156.
>
> [2] Karimi, Fariba, et al. "Inferring gender from names on the web: A comparative evaluation of gender detection methods." Proceedings of the 25th International conference companion on World Wide Web. 2016.

---

### Official Review · AnonReviewer4 · 2020-10-27
**Interesting, but out of scope**

**Rating:** 3
**Confidence:** 3

**Review:**

The paper analyzes ICLR submissions and trends in the reviewing and selection process.

The paper is very interesting, and would be quite valuable for the senior members of the community to read and be familiar with. However, it does not seem to have anything to do with the ICLR topics. Hence, in this reviewer's view, it should either be privately disseminated to area chairs, or published as some form of appendix, but should not be part of the conference.

Getting into the details, it is not clear why the metric for the regression is just the mean reviewer score, not including all grades (or at least a variance). There is information in a paper that a reviewer gives a particular high/low grade to. Constructing then simulated data that uses the variation data and being surprised they don't match sounds as a potential pitfall for this analysis.

In addition, there are several biases identified in the paper. But since the AC are not identified as biased, and the papers are anonymous, it is not clear what is the mechanism suggested by the authors of how these biases manifest themselves. Is there a suggestion that CMU/Cornell/MIT have a specific way of writing papers? Do women? Or is there a suggestion that anonymity does not genuinely exist, and most reviewers have good knowledge on who are the authors of the papers they review?

---

> ### Author Response · Authors · 2020-11-24
> **We appreciate your time and feedback.**
>
> Thank you for your feedback.  We believe that this work will be of interest not only to area chairs but to the ICLR community at large.  Thus, we feel that there is no better place to showcase this work than at ICLR itself.
>
> During model construction we looked through the data carefully, and we did not find a relationship between variance of reviews and acceptance.  Moreover, we found that logistic regression on median (rather than mean) scores yields results almost identical to regression on mean scores.
>
> Regarding the blindness of the review process: It is quite common to already be familiar with a paper before being assigned to review it given that authors may give public talks about pre-prints, and many papers appear on arXiv before review.  Additionally, reviewers or ACs may look up papers they are assigned even though they are not supposed to do this.  Authors of this submission have even reviewed different versions of the same paper multiple times at different conferences, and it is our impression that this is not all that rare.  While we do not draw a causative link via our statistical methods, we find the idea that anonymity does not genuinely exist to be entirely plausible.

---

### Official Review · AnonReviewer1 · 2020-10-28
**This paper focuses on understanding and analyzing the reviewing process for a large conference such as ICLR and understand the reproducibility of the review process.**

**Rating:** 6
**Confidence:** 3

**Review:**

This paper focuses on understanding and analyzing the reviewing process for a large conference such as ICLR and understand the reproducibility of the review process through Monte Carlo simulations.  Further, the authors also aim to study the impact of factors such as institutional bias, gender and study if higher review scores ensure more number of citations.

Comments:
1. How were the gender labels produced? Was it done through a manual process? With regard to gender bias analysis, one missing type of analysis in terms of reviewers' scores is to take into consideration the arxiv/ resubmission effect. This would provide better insights.
2. The timing of this study is important and some of the findings in this paper raise concerns about the overall review process. Also, a big thanks to the authors of this work for providing the entire codebase to reproduce their results
3. "As more reviewers are added, the high level of disagreement among area chairs remains constant, while the standard error in mean scores falls slowly" --> How is the high level of disagreement quantified? or is that an assumption? And if there is high-level of disagreement, how does the logistic regression model take that into consideration?
4.  Figure 4 is really interesting in terms of showing the impact of making submissions available earlier. This raises concerns about how making papers available online is biasing the reviewers in terms of the scores provided.
5. I have concerns over the way interrater reliability is calculated in section 4. The reviewers are randomized and this may not lead to the right number of samples per review and may affect the interrater reliability. Also, the assumptions mentioned by the authors seems to be wrong for this process.

 Suggestions:
1. "then" should be corrected to "the" in the last line of the first paragraph of section 2.
2. Can the mention of NIPS conference be changed to Neurips conference even though the change only happened in 2018?

---

> ### Author Response · Authors · 2020-11-24
> **We appreciate your feedback and corrections.**
>
> Thank you for your feedback.  We address each of your comments below.
>
> (1)  Gender labelling was done manually as described in the paper.   For each author, we visit their web page (if possible) and determine their gender by their self-selected pronouns on their web page or CV.  When this is not available we assign a label based on the canonical gender of their name. See our comments to reviewer 4 for more details on why this process was chosen for our study.  Regarding an analysis which controls for arXiv presence, we tried doing this, but there were not enough papers with female authors to further break those papers down into temporal bins and still obtain statistically significant results.
>
> (3)  We apologize for including the subjective modifier “high”.  We have revised the paper to remove this subjective judgement about the level of disagreement.  The simulation results use a logistic regression model that predicts the likelihood of a paper being accepted by an AC given its mean scores.  When this model outputs a number close to 0 or 1, the level of agreement among ACs is high, and when it outputs a number close to 0.5 agreement is low.  We quantify the level of agreement among ACs in the paper by looking at the reproducibility of decisions when the number of reviewers is infinity; in this case, only the agreement/disagreement among ACs is reflected in the simulations.  We found that, for an average accepted paper, a randomly chosen AC is 75% likely to agree with the decision to accept.
>
> (5)  Thank you for pointing this out.  We have re-calculated the values of Krippendorff’s alpha, and updated our description.  The same trend remains.
>
> Regarding typographic errors, thank you for pointing these out.  We have now corrected these.

---

### Official Review · AnonReviewer3 · 2020-10-28
**Interesting point of view with room to improve.**

**Rating:** 5
**Confidence:** 3

**Review:**

First of all I want to thank the author(s) to notice and study the quality of reviews in ICLR.

Quality:

The authors put a lot of effort in collecting and cleaning the data, which is not easy. The authors did thorough statistical analysis over the data and considered many possible pitfalls. I still have some questions but I think it's of good value for the community to see and discuss the paper in the conference.

Clarity:

The paper is well written and easy to follow.

Originality:

Good original thinking.

Significance:

The analysis and conclusion is of good value for the ICLR community. Great significance.

I gave the rating of marginally below acceptance threshold. Why not higher? I think there are some fundamental questions that need more clarification from the authors (4,5, 6,7,10 and 12 in the detailed comments). But to be clear I still think there's great value for it to be accepted given the topic, and if the authors can answer the questions in the comments.

Detailed comments:

1) Page 1, intro: What do you mean by "censorship" here?
2) Page 1 last paragraph. I think the reviewers have no knowledge of author's gender, given the double blind review process? I know you might be able to dig it through by searching arxiv ,but doubt anyone would do that to see the gender.
3) Page 2, data set. It's better to explain the size of data clearly, is it 2560 papers or 5569 papers?
4) Page 3. the simulation. Given the whole simulation set up, I think the key factor is the variation of reviewer score and AC's decision noise. Why don't we just look at those directly? Simulation gives a more intuitive number (re-acceptance rate) but with more noise injected in.
5) Page 3, I'm not a statistician, but short research on Hosmer-Lemeshow test suggests that it's not a good way to exam the quality of LR model. Could you also show the x-validation precision/recall of the prediction? AC might take other factors into consideration, such as the content of review comments, reviewer's confidence and background, or the order of papers (esp the ones on the border of a/r).
6) Page 3, If the same 2020 logistic regress model was used, then the result only shows that the variance of review score increased over time. Have you considered the change in number of papers and number of reviewers over the years?
7) Page 4, Sec 3.3. Number of citation increases exponentially over time. simply dividing by time will punish papers published sooner.
8) Page 5, figure 4, could you add the papers that never before seen online too?
9) Page 6, paragraph 4, is it 0.15 out of 10? seems a trivial amount.
10) Page 7, reputation, Is this a correlation or causality? It could be that papers from authors with better reputation did have an edge in quality. Remember ACs have access to the review notes and the paper themselves too.
11) Page 7, second last paragraph.  We can't mix woman % in the US with ICLR which is a global venue. Global % could be lower.
12) Page 8 Conclusion. This is a weak point, Conclusion is not supported by the content of the paper, but more like a speculation.

---

> ### Author Response · Authors · 2020-11-24
> **Thank you for your insightful feedback.**
>
> Thank you for your insight.  We address each of your comments below.
>
> 1.  Some authors feel that papers may be rejected because they present competition for other authors, because of personal conflicts, or other reasons unrelated to paper “quality.”  We agree that “censorship” is both vague and overly harsh, so we have changed it to “a process orthogonal to meritocracy”.
>
> 2.  We are not suggesting that reviewers and ACs look up papers on arXiv specifically to find out the authors’ genders, but any reviewer who knows the author list may coincidentally know the authors’ genders.  We would point out that there are many ways that the review process is not blind.  It is fairly common for a reviewer to be assigned a paper that they are already familiar with from arXiv (or a workshop venue), or from a talk that they’ve seen.  Also, while this is officially prohibited by reviewing guidelines, we suspect that many reviewers search arXiv for papers regardless.  That being said, we suspect that much of the difference in scores that we observe is the result of the well-documented “achievement gap” between men and women in the sciences (the citations at the top of section 5.4).  Women tend to have smaller peer groups than men in computer science (and in scientific fields at large) that result in worse learning outcomes, lower productivity levels, or may prompt women to leave the field all together.  This effect might be particularly strong at ICLR, where roughly 1 out of 10 first or last authors are male.
>
> 3.  Thank you for pointing this out.  We have clarified this in our current version.  There are a total of 5569 papers over 4 years. ICLR 2020 had 2560 submissions, ICLR 2019 had 1565, ICLR 2018 had 960, and ICLR 2017 had 490.
>
> 4.  We agree that the variation in scores and AC decisions is useful.  We think our results already contain this, and we agree that we should feature these numbers more prominently.  The ANOVA measures the variation between reviewer scores when controlling for paper “quality” (i.e., the expected paper score).  The ANOVA model finds that reviewer scores have a standard deviation of 1.72, and this is reported in the main body of the paper.  Quantifying randomness in the AC judgements is more complex, since AC judgements depend strongly on scores; a low scoring paper will nearly always be rejected (no significant variation in AC judgment), and a high scoring paper will always be accepted.  Likewise, assuming a continuous model, there exists a paper score for which an AC will have a 50% chance of accepting a paper, and so reporting the worst-case randomness is not particularly interesting.  Rather, we address the problem in Figure 2, where we show that if the number of reviewers is infinity (i.e., we completely remove the randomness in reviewer scores, and all randomness comes from the AC) then the re-acceptance rate is 75%.  This means that, for an average accepted paper, a randomly chosen AC has a 75% chance of agreeing that the paper should be accepted.  We have now added figures depicting the reproducibility of rejections and of all papers overall.
>
> 5. It’s true that there are more sophisticated goodness of fit tests.  We presented HL because it is the most standard method.  HL’s main weakness is that it is vulnerable to overfitting.  In our case, the model has very few parameters (<20 for all LR models in this paper), the dataset is large (>2500 when using 2020 data alone), and we were not concerned about the overfitting issue.  Nonetheless, we have now run Stukel’s test, a more recent goodness-of-fit test for logistic regression, and we have added this to the appendix.
>
> 6.  The reproducibility study used a unique regression model for each year trained only on that year’s data.  We did not recycle the 2020 AC model for use in other years.  In the description of the Monte-Carlo method, we used the year 2020 as an example, but we did not mean to imply that 2020 was used for every year.  Thanks for pointing out this weakness in the presentation;  we have now clarified this in our current version.

---

> > ### Author Response · Authors · 2020-11-24
> > **Thank you for your insightful feedback.**
> >
> > 7.  We spent a lot of time looking at citation data, and we find that the “exponential” increase in citation counts only holds for a minority of extremely popular papers, and the vast majority of papers exhibit a roughly linear citation trend over time.  Note:  In section 5.3, we assume an exponential growth model for (and use a log transform to normalize) the number of citations for last authors.  This is because last authors accumulate papers over their career and so we observe a clear acceleration of their citation rates over time.  Moreover, last authors are likely to have authored at least one of these extremely popular papers with exponential citation rate, and often their total citation number is dominated by such papers.
> > Just to be sure, we have added a section to the appendix in which we assume exponential growth of paper citations, and we take the log of citation count before dividing by time. Computing spearman correlation coefficients yields a similarly small relationship between paper score and impact within each decision category.
> >
> > 8.  Thank you for this suggestion.  We have added two new figures to the appendix showing results separated in the way you requested.  The left-most blue bar in the new graphs is the acceptance rate of papers that did not appear in any online venue before the submission date. The new figures still indicate that resubmitted papers have higher acceptance rates, and papers that can't be found online during the review process have even lower acceptance rates.
> >
> > 9.  Keep in mind that for many borderline papers, one reviewer boosting their score by a single point may cause an otherwise rejected paper to be accepted.  For perspective, if a paper has three reviewers, then one reviewer increasing their score by one point corresponds to a boost of 0.33 points to the mean score.  In our further studies, we observe that a handful of institutions in the top 10 exhibit a statistically significant boost. When examining only CMU, MIT, and Cornell, we feel that boosts of 0.31, 0.31, and 0.58 respectively are noteworthy.
> >
> > 10.  Thank you for pointing this out.  We agree with your point, and we don’t want to mislead anyone into thinking we have established causality.  We should not have used the term “impact” (but rather “correlation” or “relationship”) and have updated our draft to reflect this ambiguity.
> >
> > 11.  Thank you for catching this discrepancy.  We have now updated this section of our paper to measure the proportion of ICLR submissions with female first/last authors only out of US universities.
> >
> > 12.  In the main body of our paper, we try to keep the focus solely on objective content.  We felt that the conclusion section of the paper was an appropriate place to discuss subjective opinions.  While we do make several speculations, we tried our best to make it clear that this section is subjective (indeed, we use the words “speculate” and “interpret” to describe our conclusions).  We take your comment seriously, and we will discuss it among the authors and take it into consideration when making future revisions.

---

### Decision · Program_Chairs · 2021-01-07
**Final Decision**

**Decision:**

Reject

**Comment:**

Reviewers appreciated the care and substantial effort that went into the paper, for instance:
AR3) I think it's of good value for the community to see and discuss the paper in the conference.
AR4) would be quite valuable for the senior members of the community to read and be familiar with.

The main argument for rejection is the the analysis done in the paper is not typical of ICLR research.  Arguably, the paper could fall under the topic "societal considerations of representation learning including fairness, safety, privacy", but this does not apply because the subject of analysis is the conference ICLR, not representation learning.   I support this argument.

The reviewers posed a good number of questions and issues with the paper, and largely these were addressed well by the authors.  In some cases they addressed the issues properly, and others they argued their case.  For instance AR2 says  "think the ACs decision process is too simplified" and the response summed up as "our ability to do multi-factor studies is limited by the size of our dataset".

An important one of these discussions is as follows:
AR4)  But since the AC are not identified as biased, and the papers are anonymous, it is not clear what is the mechanism suggested by the authors of how these biases manifest themselves.
Authors)  <extensive points>  .... we find the idea that anonymity does not genuinely exist to be entirely plausible.
I would argue that neither party can claim to have won this argument, and I am not really sure how it can be resolved.  Fortunately, though, no evidence for gender bias in ACs was found.

In conclusion, the paper is not topical to ICLR material, and the reviewer consensus is Reject.  However, the paper is both valuable and interesting to the community, and it has seen substantial improvement through the review process and a lot of the issues defended well.

The paper should be brought to the attention of the various committees and made available somehow at the conference and acknowledged as a useful publication.